# A Model Assessment of the Occurrence and Reactivity of the Nitrating/Nitrosating Agent Nitrogen Dioxide (^•^NO_2_) in Sunlit Natural Waters

**DOI:** 10.3390/molecules27154855

**Published:** 2022-07-29

**Authors:** Davide Vione

**Affiliations:** Dipartimento di Chimica, Università degli Studi di Torino, Via Pietro Giuria 5, 10125 Torino, Italy; davide.vione@unito.it; Tel.: +39-011-6705296

**Keywords:** environmental chemistry, photochemistry, indirect photolysis, photonitration, reactive nitrogen species

## Abstract

Nitrogen dioxide (^•^NO_2_) is produced in sunlit natural surface waters by the direct photolysis of nitrate, together with ^•^OH, and upon the oxidation of nitrite by ^•^OH itself. ^•^NO_2_ is mainly scavenged by dissolved organic matter, and here, it is shown that ^•^NO_2_ levels in sunlit surface waters are enhanced by high concentrations of nitrate and nitrite, and depressed by high values of the dissolved organic carbon. The dimer of nitrogen dioxide (N_2_O_4_) is also formed in the pathway of ^•^NO_2_ hydrolysis, but with a very low concentration, i.e., several orders of magnitude below ^•^NO_2_, and even below ^•^OH. Therefore, at most, N_2_O_4_ would only be involved in the transformation (nitration/nitrosation) of electron-poor compounds, which would not react with ^•^NO_2_. Although it is known that nitrite oxidation by CO_3_^•^^−^ in high-alkalinity surface waters gives a minor-to-negligible contribution to ^•^NO_2_ formation, it is shown here that NO_2_^−^ oxidation by Br_2_^•^^−^ can be a significant source of ^•^NO_2_ in saline waters (saltwater, brackish waters, seawater, and brines), which offsets the scavenging of ^•^OH by bromide. As an example, the anti-oxidant tripeptide glutathione undergoes nitrosation by ^•^NO_2_ preferentially in saltwater, thanks to the inhibition of the degradation of glutathione itself by ^•^OH, which is scavenged by bromide in saltwater. The enhancement of ^•^NO_2_ reactions in saltwater could explain the literature findings, that several phenolic nitroderivatives are formed in shallow (i.e., thoroughly sunlit) and brackish lagoons in the Rhône river delta (S. France), and that the laboratory irradiation of phenol-spiked seawater yields nitrophenols in a significant amount.

## 1. Introduction

Photochemical reactions are important processes in sunlit natural surface waters. They play a significant role in the transformation of biorecalcitrant pollutants, and of some natural compounds, and often result in decontamination. However, sometimes photochemistry yields secondary contaminants, which may be more harmful than the parent molecules [1,2]. Phototransformation by direct photolysis is operational for those compounds, which absorb sunlight, and get degraded as a consequence, because they have a non-nil quantum yield of direct photolysis [3,4]. Conversely, indirect photochemistry is the transformation of dissolved compounds upon reaction with the so-called photochemically produced reactive intermediates (PPRIs), independent of sunlight absorption by the molecules that get transformed [5,6]. The main PPRIs in natural surface waters are the hydroxyl (^•^OH) and carbonate (CO_3_^•−^) radicals, the excited triplet states of chromophoric dissolved organic matter (^3^CDOM*, where CDOM is the chromophoric fraction of the dissolved organic matter, DOM), and singlet oxygen (^1^O_2_) [7]. PPRIs are produced upon sunlight absorption by photosensitisers, i.e., naturally occurring compounds such as nitrate and nitrite (^•^OH sources), as well as CDOM (source of ^3^CDOM*, ^1^O_2_, and ^•^OH) [8,9,10]. Moreover, CO_3_^•−^ is generated upon oxidation of HCO_3_^−^/CO_3_^2^^−^ by ^•^OH, and of CO_3_^2^^−^ by ^3^CDOM* [11,12]. After being produced, PPRIs can be quickly quenched/scavenged by DOM, either chromophoric or not. In particular, DOM scavenges ^•^OH and CO_3_^•−^, but it scavenges ^3^CDOM*/^1^O_2_ only to a very minor extent. Other important scavengers/quenchers are inorganic carbon, i.e., HCO_3_^−^/CO_3_^2^^−^ (for ^•^OH), dissolved oxygen (for ^3^CDOM*, to yield ^1^O_2_), and collision with the water solvent (for ^1^O_2_) [6]. A schematic of the main processes involving photosensitizers, PPRIs’ production, their scavenging/quenching, and interaction with pollutants is provided in Figure 1.

In addition to ^•^OH, CO_3_^•−^, ^3^CDOM*, and ^1^O_2_, there are other PPRIs that are either lesser known at the moment, e.g., longer-lived species such as superoxide and organic peroxyl radicals [13,14], or have the potential to produce harmful secondary contaminants to a higher extent compared to the PPRIs depicted in Figure 1. Examples are the dibromine (or dibromide) radical (Br_2_^•−^), which is mainly produced upon ^•^OH scavenging by bromide in saltwater [15] and is an effective brominating agent, especially for phenols [16], as well as nitrogen dioxide (^•^NO_2_). The latter is mostly generated by nitrate photolysis together with ^•^OH, and by the oxidation of nitrite by ^•^OH itself [8,17]. Other ^•^NO_2_ formation processes, the environmental importance of which is still to be conclusively elucidated, consist in the oxidation of nitrite by either ^3^CDOM* [18], or irradiated Fe(III) oxides [19].

^•^NO_2_ is a nitrating/nitrosating agent, which is involved in the production of toxic nitroderivatives from aromatic compounds [20], as well as of toxic and, possibly, mutagenic/carcinogenic nitrosoderivatives from amines and amino acids [21,22]. ^•^NO_2_ is not the only possible nitrating agent in an aqueous solution, but it is probably the most likely one to be involved in photonitration processes in circumneutral conditions. Actually, studies on the (photo)nitration pathways of phenols and other aromatic compounds have found a plethora of nitrating agents (^•^NO_2_, HNO_2_, HOONO, H_2_OONO^+^, and possibly also N_2_O_4_), but most of them tend to be operational at an acidic pH only [20,23]. Indeed, although in (often acidic) atmospheric waters the actual (photo)nitration pathways, and the reactive species involved, may still be open to debate [24], in the case of natural surface waters, ^•^NO_2_ is more likely to play a substantial role [25].

In this work, a model approach based on (photo)reaction kinetics and a steady-state approximation is applied to assess the conditions that would most favor the occurrence of ^•^NO_2_ (as well as its dimer, N_2_O_4_) in sunlit natural surface waters. Model findings help explain why, so far, literature reports about environmental photonitration processes in natural surface waters have focused on shallow and brackish lagoons, near the sea [26,27,28]. 

## 2. Kinetic Model Development

Nitrogen dioxide is produced in natural surface waters upon the photolysis of nitrate, and upon the oxidation of nitrite by ^•^OH [29]:(1)NO3−+hν+H+ → •OH+ •NO2 [RO•HNO3−]
NO_2_^−^ + ^•^OH → ^•^NO_2_ + OH^−^   [*k*_2_ = 1×10^10^ M^−1^ s^−1^](2)

The quenching/scavenging processes of ^•^NO_2_ in an aqueous solution involve dimerization into dinitrogen tetroxide (N_2_O_4_), another potential nitrating agent, followed by the hydrolysis of the latter, as well as an ^•^NO_2_ reaction with the dissolved natural organic matter (DOM) [8]:2 ^•^NO_2_ ⇆ N_2_O_4_  [*k*_3_ = 4.5 × 10^8^ M^−1^ s^−1^; *k*_−3_ = 6.9 × 10^3^ s^−1^](3)
^•^NO_2_ + DOM → Products   [*k*_4_](4)
N_2_O_4_ + H_2_O → NO_3_^−^ + NO_2_^−^ + 2 H^+^  [*k*_5_ = 1 × 10^3^ s^−1^](5)

The value of *k*_4_ is still to be conclusively assessed. However, in the case of phenolic compounds, it is reported that *k*_4_~10^4^ M^−1^ s^−1^ [30]. Moreover, it is well known that phenolic moieties are ubiquitous in DOM. When considering the dissolved organic carbon (DOC) contents of phenols [31], one may assume *k*_4_ = 0.14 L mg_C_^−1^ s^−1^.

By assuming the formation rate of ^•^NO_2_ (RN•O2), as RN•O2=RO•HNO3−+k2×[O•H]×[NO2−], and applying the steady-state approximation to both ^•^NO_2_ and N_2_O_4_ (i.e., $d[NO_{2}]/dt=0$, and $d[N_{2}O_{4}]/dt=0$), from reactions (1–5), one gets the following:(6){[N•O2]=(k−3+k5)−k4 DOC +(k4 DOC)2+8 k3k5RN•O2(k−3+k5)−14k3k5[N2O4]=k3 [N•O2]2k−3+k5

Preliminary calculations showed that reaction (4) would strongly prevail over (3,5) as an ^•^NO_2_ sink. The rate difference between the two kinds of processes is so big that the conclusion would not change even if *k*_4_ = 0.14 L mg_C_^−1^ s^−1^ turned out to be a generously high estimate for the reaction rate constant between ^•^NO_2_ and DOM. Therefore, when neglecting reactions (3,5) as ^•^NO_2_ sinks, one gets a considerable simplification for (6):(7){[N•O2]=RN•O2k4 DOC[N2O4]=k3 [N•O2]2k−3+k5

Again, the same preliminary calculations suggested that (7) approximates (6) to better than 4‰, in a wide variety of conditions that are significant for surface waters. Based on the above reactions, it appears that the main water components that are expected to impact [^•^NO_2_] and [N_2_O_4_] the most are the following:DOC (dissolved organic carbon), when considering that DOM is the main ^•^NO_2_ scavenger [18], and that it also scavenges ^•^OH [32], which plays a key role in the oxidation of NO_2_^−^ to ^•^NO_2_ [19];Inorganic carbon (HCO_3_^−^ and CO_3_^2−^), because it enhances nitrate photolysis due to a solvent cage effect (see Figure 2) [33], but also acts as an ^•^OH scavenger [32]. Interestingly, there is evidence that nitrite oxidation by CO_3_^•−^ does not contribute to ^•^NO_2_ formation significantly [18];Bromide (Br^−^), as a major ^•^OH scavenger in saltwater and seawater [15,32].

The effect of different water components on [^•^NO_2_] and [N_2_O_4_] was modeled by assessing the values of RO•HNO3− and [^•^OH], which are needed to calculate RN•O2, by means of the APEX software (Apex Srl; Modena, Italy). APEX (Aqueous Photochemistry of Environmentally occurring Xenobiotics) computes steady-state concentrations of reactive transient species (^•^OH, CO_3_^−•^, ^1^O_2_, and ^3^CDOM*) as a function of water chemistry, depth, and seasonal sunlight irradiance [34,35,36]. In this case, typical irradiance conditions for APEX were assumed: 22 W m^−2^ UV irradiance, i.e., 290–400 nm, which corresponds to fair weather 15 July at 45° N latitude, at 9 a.m. or 3 p.m., solar time. By doing so, it was possible to compute RN•O2=RO•HNO3−+k2×[O•H]×[NO2−], as well as the steady-state [^•^NO_2_] and [N_2_O_4_], as per Equation (6) or Equation (7).

## 3. Results and Discussion

### 3.1. Effect of Water Parameters on ^•^NO_2_ Formation and Occurrence

First of all, the environmental occurrence of ^•^NO_2_ and N_2_O_4_ was assessed and compared with the other transient species (PPRIs). The effect of the two main ^•^NO_2_/N_2_O_4_ sources (NO_3_^−^ and NO_2_^−^) was modeled first, to check whether or not the steady-state [^•^NO_2_] and [N_2_O_4_] were linearly dependent on [NO_3_^−^] and [NO_2_^−^]. The steady-state concentrations of the reactive transient species were thus calculated as a function of the concentration values of nitrate and nitrite, letting all the other parameters (DOC, HCO_3_^−^, CO_3_^2−^, and water depth) constant. Calculation results are shown in Figure 3 for ^•^NO_2_, N_2_O_4_, ^•^OH, and CO_3_^•−^. It is suggested that the concentration values of all these transient species increased with increasing nitrate and nitrite. At the same time, [^3^CDOM*] and [^1^O_2_] were constant at around 10^−16^ M in all conditions. These results are easily explained by the fact that nitrate and nitrite are both ^•^OH sources, and ^•^OH plays an important role in the production of both ^•^NO_2_ from NO_2_^−^, and CO_3_^−•^ from HCO_3_^−^/CO_3_^2^^−^ [6]. Moreover, the production of ^•^OH by NO_3_^−^ is closely associated with the formation of ^•^NO_2_ (reaction (1)).

Increasing the [NO_2_^−^] leads to increasing [^•^OH] and, considering that RN•O2NO2−=k2×[O•H]×[NO2−], there is a higher-than-linear effect of [NO_2_^−^] on the formation rate of ^•^NO_2_ by NO_2_^−^ itself. Moreover, because [N_2_O_4_] ∝ [^•^NO_2_]^2^ (Equation (7)), one explains the faster-than-linear increase of [N_2_O_4_] with the increasing nitrite, reported in Figure 3b.

An interesting issue is that [N_2_O_4_] is 6–7 orders of magnitude or more lower than [^•^NO_2_], and it is even lower than [^•^OH]. This means that, for N_2_O_4_ to be a competitive nitrating agent with ^•^NO_2_, it should react much faster than ^•^NO_2_ with organic compounds. This is highly unlikely, with the possible exception of very electron-poor aromatics, which would not react with ^•^NO_2_ to a significant extent. Therefore, the reported finding, that the nitration of some pyridine derivatives involves N_2_O_4_ as a nitrating agent [37], looks more like an exception rather than a typical event.

Additionally, the steady-state [CO_3_^•−^] follows the same trend as [^•^OH], because ^•^OH is the main CO_3_^•−^ source through the oxidation of HCO_3_^−^/CO_3_^2−^ [11] (Figure 3d). Interestingly, there is evidence that NO_2_^−^ oxidation by CO_3_^•−^ does not contribute much to ^•^NO_2_ production [18].

As reported in Figure 4, nitrate photolysis (reaction (1)) would prevail as an ^•^NO_2_ source over nitrate oxidation by ^•^OH (reaction (2)) in the vast majority of the conditions depicted in Figure 3. For nitrite oxidation to prevail, one needs [NO_3_^−^] < 10 [NO_2_^−^]. In contrast, in natural waters, it is often found that [NO_3_^−^]~10^2^ [NO_2_^−^] [38]. Note that NO_3_^−^ and NO_2_^−^ play comparable roles as ^•^OH sources when [NO_3_^−^]~10^2^ [NO_2_^−^] [34].

The possible role of HCO_3_^−^ and CO_3_^2−^ in ^•^NO_2_ formation is depicted in Figure 5, where conditions were chosen so that nitrate and nitrite contributed equally to ^•^NO_2_ generation (11 µM NO_3_^−^, 0.85 µM NO_2_^−^). It appears that inorganic carbon species would play a limited and slightly negative role towards the occurrence of ^•^NO_2_, presumably because their role as ^•^OH scavengers in the solution bulk prevails over the solvent cage effect that enhances nitrate photolysis by inhibiting the geminate recombination of O^•−^ and ^•^NO_2_ (see Figure 2 for such a solvent cage effect).

DOM as an ^•^OH scavenger is expected to inhibit ^•^NO_2_ formation by quenching reaction (2), that is, the nitrite route to ^•^NO_2_. Moreover, the chromophoric moieties within DOM (i.e., CDOM) compete with radiation absorption by both nitrate and nitrite, thereby inhibiting their photolysis [34]. As a consequence, the photogeneration of ^•^NO_2_ by nitrate, and that of ^•^OH by nitrate and nitrite, are both inhibited by CDOM. Last but not least, reaction with DOM is the main scavenging process for ^•^NO_2_. For all of these reasons, the overall DOM effect to decrease [^•^NO_2_] is very important, as shown in Figure 6a.

Figure 6b,c additionally shows the decreasing formation rates of ^•^NO_2_ from nitrate and nitrite with increasing DOC, mostly due to competition for irradiance by CDOM (both cases), and ^•^OH scavenging by DOM (^•^NO_2_ formation from NO_2_^−^).

The results shown in Figure 6a were obtained under the hypothesis that (C)DOM mostly operates as an irradiance competitor and ^•^OH scavenger. However, there is also the possibility that ^3^CDOM* oxidizes NO_2_^−^ to ^•^NO_2_, and such a process is expected to contribute to ^•^NO_2_ production to a higher extent when the DOC is higher. There is evidence that a rate constant around 10^9^ M^−1^ s^−1^ would be an upper limit for the reaction kinetics between ^3^CDOM* and NO_2_^−^ [18]. That would be an upper limit, as well, for the ability of ^3^CDOM* to offset the inhibition effects by (C)DOM, shown in Figure 6a. As reported in Figure 6d (compare with Figure 6a), [^•^NO_2_] would undergo almost negligible variations, even when considering such an upper-limit ^3^CDOM* contribution. Therefore, the overall role of (C)DOM towards the occurrence of [^•^NO_2_] is strongly negative. 

The bromide anion plays a minor role as an ^•^OH scavenger in most freshwaters, but its importance increases considerably with increasing salinity, until it becomes the main ^•^OH scavenger in seawater, where [Br^−^]~0.8 mM [6,15]. Indeed, the second-order reaction rate constant between Br^−^ and ^•^OH is 1.1 × 10^10^ M^−^^1^ s^−^^1^, while the reaction rate constant between ^•^OH and DOM is in the (2–5) × 10^4^ L mg_C_^−^^1^ s^−^^1^ range [6,15]. This means that one would need DOC = 160–400 mg_C_ L^−^^1^, which is hardly reasonable for a water matrix, to scavenge ^•^OH at a comparable level as 0.8 mM Br^−^.

It is shown in Figure 7 that despite the important role of Br^−^ as an ^•^OH scavenger in saltwater and seawater, increasing [Br^−^] decreases [^•^NO_2_] only to a rather limited extent, because Br^−^ is only able to inhibit the nitrite pathway to ^•^NO_2_ (reaction (2)). Indeed, differently from (C)DOM, Br^−^ is not able to inhibit nitrate or nitrite photolysis, or to directly scavenge ^•^NO_2_. Figure 7 also shows that doubling the DOC from 1 to 2 mg_C_ L^−1^ has a far more important effect on [^•^NO_2_] than an increase in [Br^−^] by an order of magnitude.

The reaction between Br^−^ and ^•^OH yields Br^•^, and then Br_2_^•−^ upon further reaction with Br^−^ [15]. Br_2_^•−^ is able to oxidize NO_2_^−^ to ^•^NO_2_ (reaction rate constant of 2×10^7^ M^−1^ s^−1^) [39], thereby contributing to ^•^NO_2_ generation. The radical Br_2_^•−^ can dimerize to form Br^−^ and Br_2_, with a rate constant of 1.8 × 10^9^ M^−1^ s^−1^ [39], but the main quenching reaction of Br_2_^•−^ in natural waters is scavenging by DOM, with an estimated rate constant of 3 × 10^2^ L mg_C_^−1^ s^−1^ [40]. A schematic of the mentioned processes involving Br_2_^•−^ is provided in Figure 8. As shown in Figure 7, when taking into account the oxidation of NO_2_^−^ by Br_2_^•−^ (see the curve highlighted as “Br_2_^•−^” in the plot), [^•^NO_2_] would significantly increase over the levels obtained by neglecting the Br_2_^•−^ reactions. Indeed, when considering the whole process, Br^−^ does not appear to inhibit ^•^NO_2_ occurrence to a significant extent. On the one side, these results show that ^•^NO_2_ is more sensitive to the DOC than to Br^−^. However, Br^−^ is able to enhance the formation of ^•^NO_2_ through Br_2_^•−^, presumably because Br^−^ acts as an effective electron shuttle between ^•^OH and NO_2_^−^ (see Figure 8). Indeed, the reaction rate constant between Br_2_^•−^ and DOM is a couple of orders of magnitude lower than the rate constant between ^•^OH and DOM [6,40].

### 3.2. Role of ^•^NO_2_ in the Transformation/Nitrosation of Glutathione (GSH)

GSH is a tripeptide that plays an important role as an antioxidant in living organisms [41,42]. GSH does not undergo direct photolysis because it does not absorb sunlight [43], but it is photochemically degraded by ^•^OH in sunlit natural waters (second-order reaction rate constant of 3.5 × 10^9^ M^−1^ s^−1^) and by ^3^CDOM* (8 × 10^7^ M^−1^ s^−1^). In saltwater, some role is also played by Br_2_^•−^ (2 × 10^8^ M^−1^ s^−1^) [44]. ^•^NO_2_ reacts with GSH by transforming it into nitroso-GSH, with a second-order reaction rate constant in the order of 10^7^ M^−1^ s^−1^ [45].

In the case of freshwaters, it is suggested in Figure 9a that ^•^NO_2_ would play a minor role in the photochemical transformation of GSH, which is dominated by ^•^OH and by ^3^CDOM*. In this circumstance, the relative role of ^•^OH decreases and that of ^3^CDOM* increases with increasing DOC, because ^•^OH is scavenged by organic matter, the chromophoric fraction of which is, vice versa, the source of ^3^CDOM*. 

The scenario gets very different in saltwater (Figure 9b), where ^•^OH is effectively scavenged by bromide, and where the ^•^OH role in GSH degradation is strongly decreased as a consequence. At the same time, the contribution of NO_2_^−^ oxidation by Br_2_^•−^ enhances the role of ^•^NO_2_ in the transformation of GSH. In seawater conditions (0.8 mM Br^−^), ^•^NO_2_ and ^3^CDOM* would be the main reactive species for GSH transformation, with their relative role depending on the DOC, which enhances ^3^CDOM* and inhibits ^•^NO_2_, and on nitrate and nitrite concentration values.

## 4. Conclusions

The nitrating and nitrosating agent ^•^NO_2_ is produced in sunlit natural waters upon nitrate photolysis and upon nitrite oxidation by ^•^OH. The nitrate process would usually prevail in typical conditions found in natural waters, except when [NO_2_^−^] > 0.1 [NO_3_^−^]. Obviously, elevated concentration values of nitrate and nitrite are very favorable to the occurrence of ^•^NO_2_. Inorganic carbon has a limited effect on the steady-state [^•^NO_2_], because small positive and negative effects offset each other, while elevated DOC is highly detrimental to the occurrence of ^•^NO_2_. Indeed, organic matter competes with nitrate and nitrite for sunlight irradiance and, therefore, for ^•^OH photoproduction. Moreover, DOM scavenges ^•^OH that is needed for NO_2_^−^ oxidation, and it also directly scavenges ^•^NO_2_.

It is suggested here that bromide occurring in saltwater and seawater would favor the degradation processes induced by ^•^NO_2_ (e.g., glutathione nitrosation) by decreasing the role of ^•^OH and by enhancing that of ^•^NO_2_: indeed, in the presence of bromide, a further source of ^•^NO_2_ is operational, which is represented by NO_2_^−^ oxidation by Br_2_^•−^. In these conditions, the couple Br^−^/Br_2_^•−^ acts as an effective electron shuttle between ^•^OH and NO_2_^−^. This latter issue might explain why the photonitration of several phenolic compounds has been observed in the brackish waters of the Rhône delta lagoons (Southern France) [26,27,28]. In these environments, the concentration of nitrate (around 50 µM [26]) is not particularly high, despite there being important impact by agricultural activities, partly because of the elevated denitrification ability of paddy fields, and partly perhaps because of the assimilation of inorganic nitrogen by algae during the summer season [38]. By comparison, these levels are just double when compared to some mountain lakes (over 2000 m asl, NW Italy, 30 km as the crow flies off the city of Torino [46]), where nitrate occurs because of atmospheric depositions, but microorganisms are not much able to consume it. They are also comparable to the nitrate levels occurring in presently oligotrophic Lago Maggiore (NW Italy [47]). In the Rhône delta lagoons, bromide would play a role in inhibiting the degradation of the parent phenols by ^•^OH, and it would allow ^•^NO_2_ to significantly contribute to the production of the nitrophenols. When also considering the ability of DOM to scavenge ^•^NO_2_, the mentioned photonitration processes are more likely to take place in lagoon water (DOC = 4–5 mg_C_ L^−^^1^) compared to the flooded rice fields (DOC around 12 mg_C_ L^−^^1^) [26]. Similarly, bromide could also play a role in the photonitration of phenol in seawater [48].

## Figures and Tables

**Figure 1 molecules-27-04855-f001:**
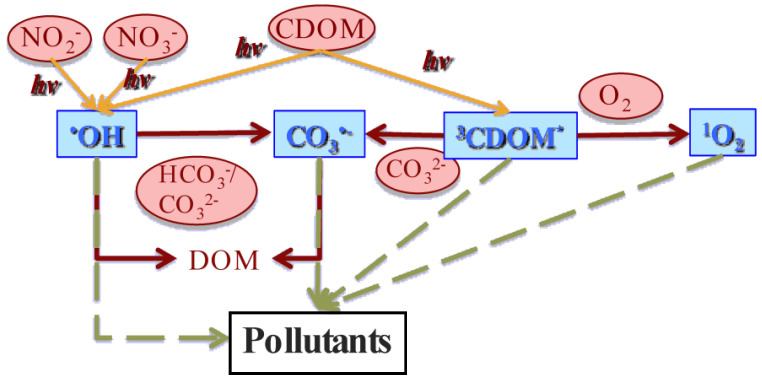
Schematic of the main processes involving photochemical production and scavenging/quenching of the main PPRIs (^•^OH, CO_3_^•−^, ^3^CDOM*, and ^1^O_2_) in natural surface waters. DOM = dissolved organic matter; CDOM = chromophoric dissolved organic matter. Circles: photosensitizers and scavengers/quenchers; rectangles: PPRIs.

**Figure 2 molecules-27-04855-f002:**
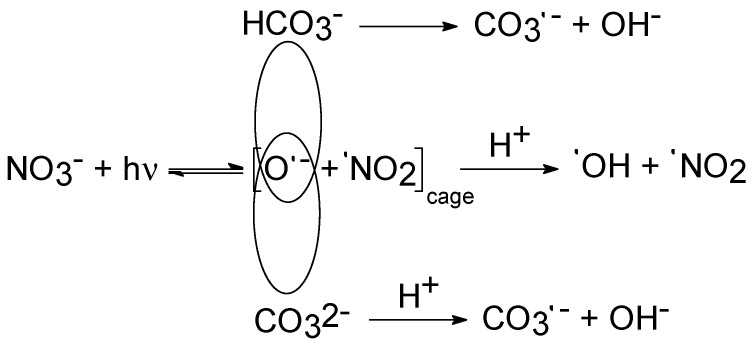
Schematic of the solvent-cage enhancement of nitrate photolysis by HCO_3_^−^ and CO_3_^2−^, which inhibits geminate recombination between the photo-fragments O^•−^ (^•^OH precursor) and ^•^NO_2_.

**Figure 3 molecules-27-04855-f003:**
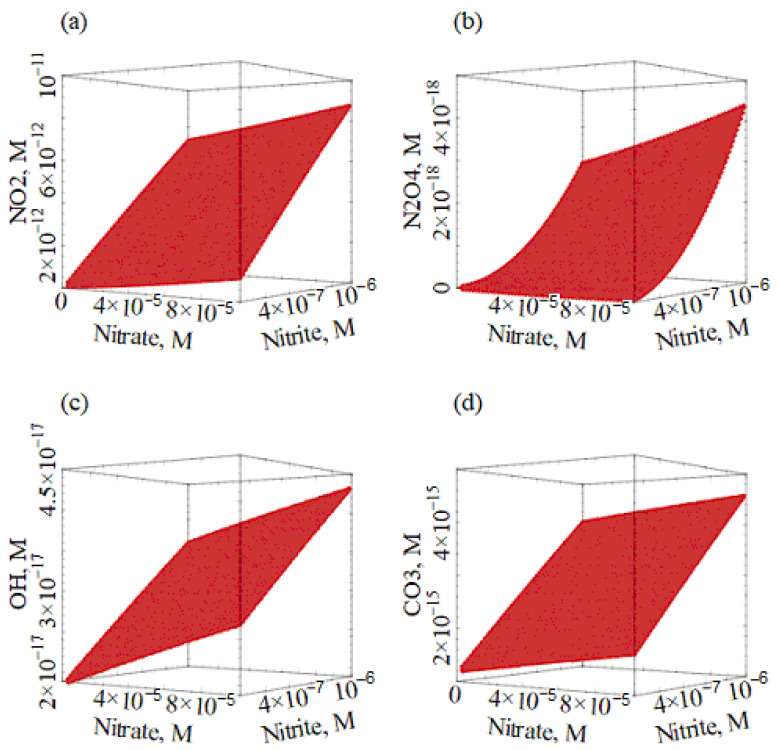
Modeled steady-state concentrations of ^•^NO_2_ (**a**), N_2_O_4_ (**b**), ^•^OH (**c**), and CO_3_^•−^ (**d**) as a function of nitrate and nitrite concentrations. Other conditions: 5 m water depth, 1 mM HCO_3_^−^, 10 µM CO_3_^2−^, 1 mg_C_ L^−1^ DOC, and 22 W m^−2^ sunlight UV irradiance, which is equivalent to fair weather, 45° N latitude 15 July, at 9 a.m. or 3 p.m.

**Figure 4 molecules-27-04855-f004:**
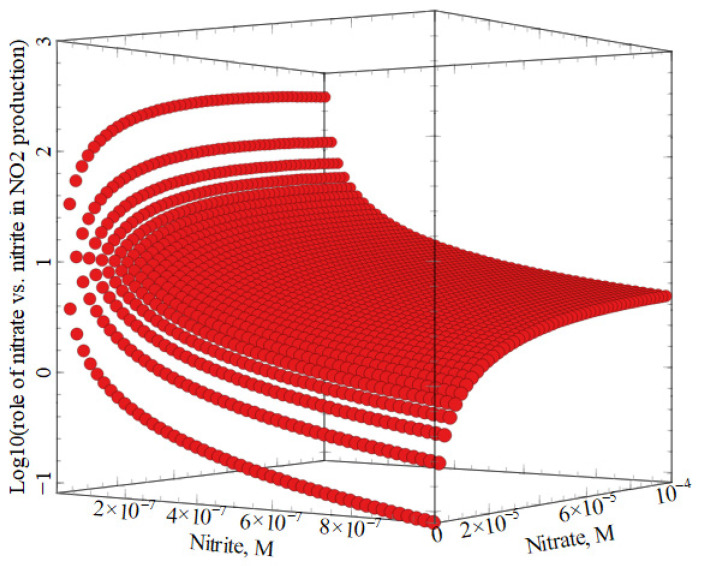
Comparison between the contributions to ^•^NO_2_ generation by nitrate photolysis (RN•O2NO3− = RO•HNO3−), and by nitrite oxidation by ^•^OH (RN•O2NO2−=k2×[O•H]×[NO2−]). Nitrate and nitrite concentrations and other conditions are the same as for Figure 3. The nitrate process prevails when the logarithm is positive (most cases), whereas the nitrite process prevails when the logarithm is negative.

**Figure 5 molecules-27-04855-f005:**
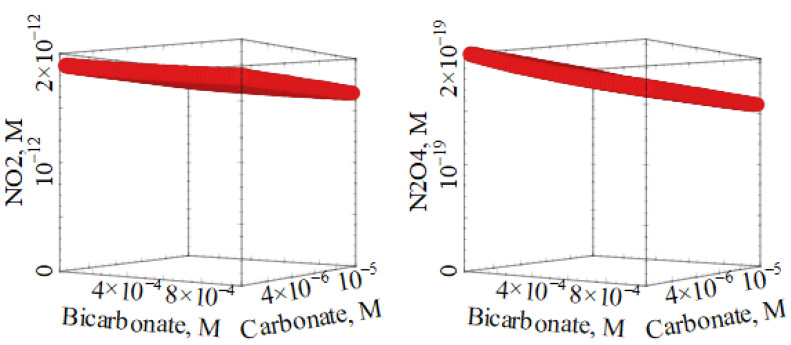
Effects of the concentration values of carbonate and bicarbonate on the steady-state concentrations of [^•^NO_2_] and [N_2_O_4_]. Other conditions: 5 m water depth, 11 µM NO_3_^−^, 0.85 µM NO_2_^−^, 1 mg_C_ L^−1^ DOC, and 22 W m^−2^ sunlight UV irradiance, which is equivalent to fair weather, 45° N latitude 15 July, at 9 a.m. or 3 p.m.

**Figure 6 molecules-27-04855-f006:**
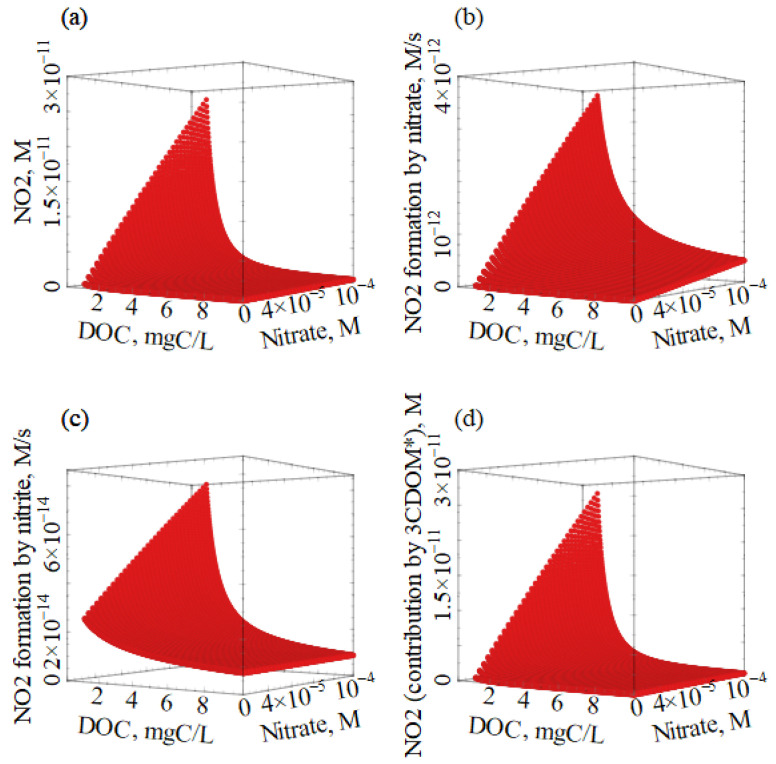
(**a**) Steady-state [^•^NO_2_], assumed to be generated by nitrate photolysis and nitrite oxidation by ^•^OH, as a function of DOC and nitrate concentration. Other conditions: 5 m water depth, [NO_2_^−^] = 10^−2^ [NO_3_^−^], 1 mg_C_ L^−1^ DOC, and 22 W m^−2^ sunlight UV irradiance (equivalent to fair weather, 45° N latitude 15 July, at 9 a.m. or 3 p.m.). In the same conditions: (**b**) ^•^NO_2_ formation rate by nitrate photolysis; (**c**) ^•^NO_2_ formation rate upon ^•^OH oxidation of nitrite, and (**d**) steady-state [^•^NO_2_], assumed to be generated by nitrate photolysis, nitrite oxidation by ^•^OH, and nitrite oxidation by ^3^CDOM*. The second-order rate constant of the latter process was assumed to be 10^9^ M^−1^ s^−1^.

**Figure 7 molecules-27-04855-f007:**
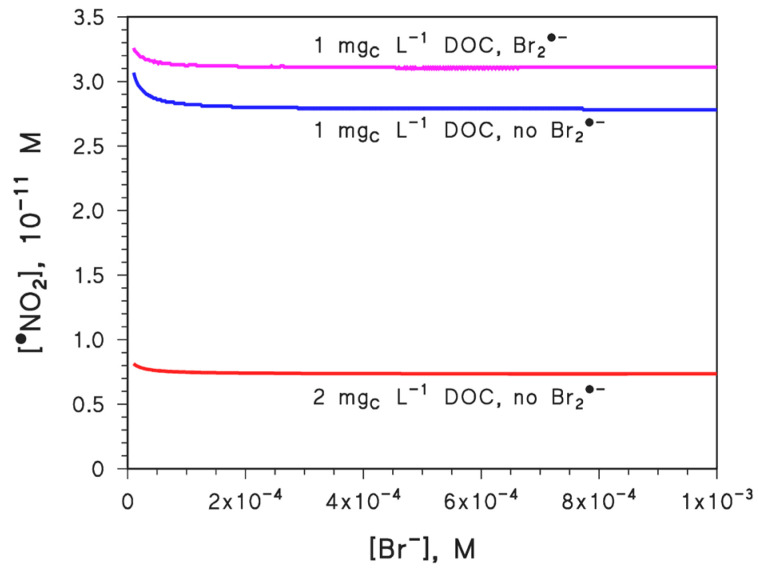
Steady-state [^•^NO_2_] as a function of bromide concentration (in typical seawater, it is [Br^−^] = 0.8 mM). The three curves differ on the value of the DOC and on whether or not the oxidation of nitrite by Br_2_^•−^ was taken into account as an ^•^NO_2_ source. Other conditions: 5 m water depth, 0.1 mM NO_3_^−^, 1 µM NO_2_^−^, 1 mM HCO_3_^−^, 10 µM CO_3_^2^^−^, and 22 W m^−2^ sunlight UV irradiance, which is equivalent to fair weather, 45°N latitude 15 July, at either 9 a.m. or 3 p.m.

**Figure 8 molecules-27-04855-f008:**
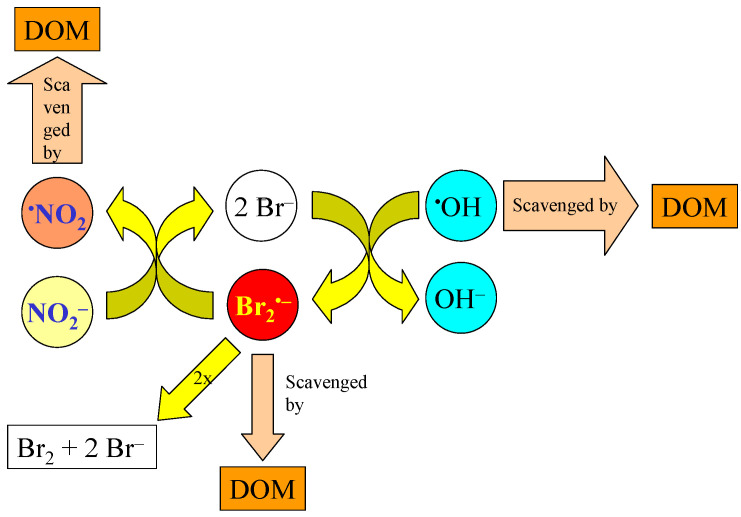
Schematic of the main processes involving Br_2_^•^^−^ in sunlit saltwater. The thickness of the scavenging arrows is intended to suggest that ^•^OH reacts with DOM much faster compared to Br_2_^•−^. Therefore, when Br^−^ outcompetes DOM as an ^•^OH scavenger, the couple Br^−^/Br_2_^•−^ acts as a very effective electron shuttle between ^•^OH and NO_2_^−^.

**Figure 9 molecules-27-04855-f009:**
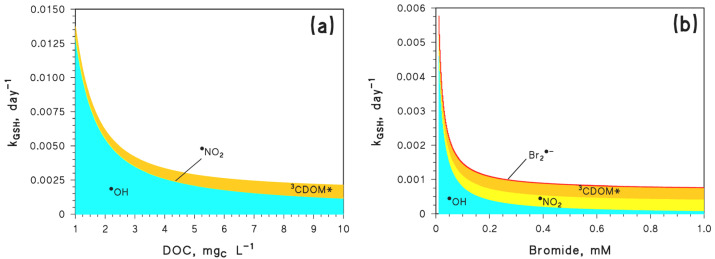
Pseudo-first order rate constants of the GSH’s indirect phototransformation, as a function of (**a**) the DOC, and (**b**) the bromide concentration. Other conditions (when the given parameter is not set to vary on the X-axis): 5 m water depth, 0.1 mM NO_3_^−^, 1 µM NO_2_^−^, 1 mM HCO_3_^−^, 10 µM CO_3_^2−^, 1 mg_C_ L^−1^ DOC, negligible Br^−^, and 22 W m^−2^ sunlight UV irradiance (equivalent to fair weather, 45°N latitude 15 July, at 9 a.m. or 3 p.m.). The color code depicts the relative importance of the different phototransformation processes.

## Data Availability

Data supporting this study can be provided by the author on request.

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
