# Peer review of "A Model Assessment of the Occurrence and Reactivity of the Nitrating/Nitrosating Agent Nitrogen Dioxide (•NO2) in Sunlit Natural Waters"

_molecules, 2022, doi:10.3390/molecules27154855_

Round 1
Reviewer 1 Report
(1) The full name of DOM should be offered when firstly mentioned in the work.
(2) The author proposed that NO2- oxidation by Br2·- can produce ·NO2 in saline waters, please explain the phenomenon.
(3) Photolysis of nitrate and oxidation of nitrite could produce ·NO2, so the concentration of ·NO2 increased with levels of nitrate and nitrite is easily to accept, so the purpose of the modelled steady-state of ·NO2 with nitrate and nitrite concentrations is what?
Reviewer 2 Report
See attached document

Round 2
Reviewer 2 Report
The manuscript has indeed been improved, but there is still room for improvements. I still have two concerns, 1) the nitrate concentration under natural, unpolluted conditions, and 2) the competition between DOM and singlet oxygen as compared to DOM and NO2 radical.
1) As far as I have been able to find out, the Rhône lagoon is quite polluted due to pollutants from agriculture and industry. Presumably, the nitrate concentration is well above that found in natural waters, so the headline's focus on natural waters leaves a wrong impression,
2) With a much lower nitrate concentration, the singlet oxygen becomes a much more important oxidant than the NO2 radical for the pollutants referred to in the introduction. This fact is neglected in Figure 1 where 1O2 does not seem to react with the pollutants.
The paper would have been much more realistic and interesting if the focus was not natural waters, but nitrate-rich waters with little DOM so that the singlet-oxygen competition could be neglected.
Author Response
I would like to thank the reviewer for his/her useful comments. In particular:
1) Despite agricultural pollution, the levels of nitrate in the Rhône lagoons are not that high. They are actually comparable to the levels found in oligotrophic or even rural environments in NW Italy, for instance. Indeed, despite agricultural inputs, nitrate is efficiently consumed by biological processes. That said, nitrate levels in the lagoons are far from extreme and they can be deemed as rather typical in surface waters. For sake of clarity, this issue is now made more explicit in page 10.
2) Singlet oxygen is often not that important in pollutant photodegradation, although with some key exceptions. In Figure 1, the arrows going from PPRIs to pollutants were there for 1O2 as well. In the revised version, the figure was slightly modified to make this issue clearer.